# Diagnosis and Treatment of Sleep Apnea in Children: A Future Perspective Is Needed

**DOI:** 10.3390/biomedicines11061708

**Published:** 2023-06-14

**Authors:** Esther Solano-Pérez, Carlota Coso, María Castillo-García, Sofía Romero-Peralta, Sonia Lopez-Monzoni, Eduardo Laviña, Irene Cano-Pumarega, Manuel Sánchez-de-la-Torre, Francisco García-Río, Olga Mediano

**Affiliations:** 1Sleep Unit, Pneumology Department, Hospital Universitario de Guadalajara, 19002 Guadalajara, Spain; 2Centro de Investigación Biomédica en Red de Enfermedades Respiratorias (CIBERES), 28029 Madrid, Spain; 3Medicine Department, Universidad de Alcalá, 28805 Madrid, Spain; 4Sleep Research Institute, 28036 Madrid, Spain; 5Sleep Unit, Pneumology Department, Hospital Universitario Ramón y Cajal, Instituto Ramón y Cajal de Investigación Sanitaria (IRYCIS), 28034 Madrid, Spain; 6Group of Precision Medicine in Chronic Diseases, University Hospital Arnau de Vilanova and Santa María, 25198 Lleida, Spain; 7Department of Nursing and Physiotherapy, Faculty of Nursing and Physiotherapy, Institut de Recerca Biomèdica de Lleida (IRBLleida), University of Lleida, 25002 Lleida, Spain; 8Respiratory Diseases Group, Respiratory Service, Hospital Universitario La Paz, IdiPAZ, 28046 Madrid, Spain; 9Pneumology Department, Hospital Universitario La Paz, IdiPAZ, 28046 Madrid, Spain; 10Faculty of Medicine, Universidad Autónoma de Madrid, 28049 Madrid, Spain

**Keywords:** sleep apnea, cardiovascular, hypoxic burden, children, diagnosis, treatment

## Abstract

Obstructive sleep apnea (OSA) in children is a prevalent, but still, today, underdiagnosed illness, which consists of repetitive episodes of upper airway obstruction during sleep with important repercussions for sleep quality. OSA has relevant consequences in the pediatric population, mainly in the metabolic, cardiovascular (CV), and neurological spheres. However, contrary to adults, advances in diagnostic and therapeutic management have been scarce in the last few years despite the increasing scientific evidence of the deleterious consequences of pediatric OSA. The problem of underdiagnosis and the lack of response to treatment in some groups make an update to the management of OSA in children necessary. Probably, the heterogeneity of OSA is not well represented by the classical clinical presentation and severity parameters (apnea/hypopnea index (AHI)), and new strategies are required. A specific and consensus definition should be established. Additionally, the role of simplified methods in the diagnosis algorithm should be considered. Finally, the search for new biomarkers for risk stratification is needed in this population. In conclusion, new paradigms based on personalized medicine should be implemented in this population.

## 1. Introduction

### 1.1. Definition and Prevalence of Obstructive Sleep Apnea in Children

Sleep-disordered breathing (SDB) occurs as a result of upper airway (UA) dysfunction (snoring and/or increased respiratory effort). It ranges from snoring to obstructive sleep apnea (OSA), depending on the degree of intermittent UA obstruction [1], and around 20% of children who snore have OSA [2].

OSA is characterized by recurrent events of partial (hypopnea) or complete (apnea) obstructions in the UA, which disrupt normal oxygenation, ventilation, and sleep patterns [1,3,4,5]. OSA in children has a clear entity with profiles that are very different from adults in terms of etiology, clinical presentation, and consequences (Figure 1). For this reason, a specific definition, diagnosis, and treatment approach is needed for this specific population.

OSA is a very frequent condition in children, with prevalence varying between 1 and 4% [6]. Although there has been an effort to increase knowledge about this entity in childhood, there is less scientific evidence than in adults. Different guidelines establish the definition of SDB and OSA in children, although the criteria are diverse and lack recent updates. The classification of OSA severity in children, through the apnea/hypopnea index-(AHI) (number of respiratory events per hour of sleep) obtained from sleep studies, is the most-commonly used parameter. Generally, an AHI of 1–3/h is accepted as the normal cutoff line for the diagnosis of OSA and is classified as follows: mild OSA if the AHI < 5/h, moderate OSA when the AHI is between 5 and 10/h, and severe OSA when the AHI > 10/h [1,3,5,7]. However, these criteria can vary depending on the guidelines, considering factors such as age, additional comorbidities, and other polysomnographic variables (presence and length of oxygen desaturations, degree of hypoventilation, sleep fragmentation, and decreased total sleep time) [7,8].

### 1.2. Etiology of OSA in Children

The etiology of childhood OSA is multifactorial, involving many risk factors, which can increase UA narrowing and collapsibility and which may contribute to the pathogenesis of OSA [8,9,10]. This includes both anatomical and neuromuscular disturbances, leading to increased airway resistance and preventing the normal function of the dilator muscles, respectively [5] (Figure 2).

The most-common risk factor is adenotonsillar hypertrophy, reaching the peak of development between 2 and 8 years [11], coinciding with the onset of OSA [12]. Nevertheless, some studies have shown a weak or no correlation between the size of the tonsils and adenoids and the severity of pediatric OSA [13,14]. Craniofacial abnormalities can also be a cause of UA narrowing: alterations of the size, position, and geometry of the mandible and the tongue [10]. These anatomical features are often found in children with craniofacial syndromes, achondroplasia, trisomy 21, Beckwith–Wiedemann syndrome, Chiari malformation, and mucopolysaccharidoses [8].

Besides anatomic factors, obesity has also been suggested as a contributor to OSA. Obese children represent a special risk factor, as the prevalence of childhood obesity is progressively increasing (5.6% in girls and 7.8% in boys) [15], also leading to an increase in the prevalence of obesity-associated morbidities including OSA [10,11,16]. This relationship is bidirectional, as OSA is known to worsen weight loss and overweight [17,18].

### 1.3. Symptoms of OSA in Children

The symptoms are classically divided into nocturnal and diurnal (Table 1). Nocturnal symptoms include snoring, witnessed apneas, gasping, oral breathing, paradoxical thoracic movements, nightmares, restless sleep, and nocturnal enuresis. Snoring is the most-common symptom, along with oral breathing. This population can also present disturbed sleep with frequent changes of position, unusual sleep positions (neck hyperextension), and nightmares [9,19]. Enuresis is another frequent symptom in OSA children related to an altered arousal response and sleep fragmentation, often being resolved when OSA is adequately treated [20].

Related to daytime symptoms, a relationship between OSA and behavioral disorders (irritability, aggressiveness, and depression), neurocognitive disorders (difficulty concentrating/learning difficulties and inattention), mood instability, and excessive daytime sleepiness has been demonstrated [5,10,21,22,23].

### 1.4. Consequences

OSA in children is associated with a number of adverse morbidities, presented as behavioral and neurocognitive disorders, growth retardation, cardiovascular (CV) diseases, and metabolic consequences, producing a negative impact on quality of life. These consequences are derived from the presence of continuous episodes of hypoxia/resaturation, sleep fragmentation, and/or changes in the intrathoracic pressure. These immediate consequences develop a cascade of intermediate mechanisms, mainly alterations in sympathetic activity, coagulation, inflammation, and oxidative stress (Figure 3).

The effects on growth are probably related to factors such as increased energy consumption and reduced production of growth hormone, whose secretion is characterized by wide and frequent peaks during sleep [9]. These adverse results may be recovered after OSA treatment, as suggested by different studies [27,28,29].

In the CV sphere, alterations in the autonomic nervous system, vasomotor tone, systemic inflammation, and atherogenesis associated with OSA are likely to induce functional disruption of the endothelium [30]. In addition, many biomarkers have been evaluated to identify this vascular damage, the C-reactive protein (CRP) being the most-studied marker. This inflammatory indicator is increased in children with OSA, with a recent study indicating that it could be reversed after treatment [31]. It has been reported that children with OSA have increased systolic and diastolic blood pressure (BP), increased BP variability, and decreased BP dipping during sleep. Observing the BP of children with OSA is essential to identifying those at risk for developing clinically significant elevated BP in adulthood [32]. There is an independent effect of OSA on cardiopulmonary function, which improves after the disorder is adequately treated [33,34,35]. Finally, these children may develop an early metabolic syndrome [10,36], this risk being six-times higher than in healthy subjects in adolescents with OSA [37]. A brief literature search of recent evidence in these spheres is described in Table 2.

New evidence regarding untreated pediatric OSA’s significant long-term morbidities affecting different organs and systems is necessary [41]. Therefore, in order to minimize the deleterious consequences related to OSA, early correct diagnosis and treatment management are mandatory. The strongest evidence shows that this population may have a significant impact on CV health in childhood and later in adulthood [30]. Tools to identify children at risk and treatment prevention indication are needed.

### 1.5. Diagnosis

The diagnosis of OSA in the pediatric population differs according to the different clinical guidelines and is described according to the Spanish [5], European [1], and American [3] guidelines in Table 3.

Although medical history and physical examination are useful to screen and determine which patients are suspected of having OSA, the sensitivity and specificity are scarce. Thus, objective sleep tests are needed. The gold standard is overnight, attended, in-laboratory polysomnography (PSG), a complex test that records neurophysiological and cardiorespiratory variables. The American Academy of Sleep Medicine (AASM) in 2007 described the rules for the scoring of sleep and respiratory events in PSG recordings [4], last upgraded in February 2023, these being different for children than adults.

However, PSG may not be readily available, so alternative diagnostic tests can be performed: daytime nap PSG, ambulatory PSG, respiratory polygraphy (RP), nocturnal oximetry, the Pediatric Sleep Questionnaire, or nocturnal video recording. The complexity and limitations of PSG entail an increase in the development and validation of alternative methods for the diagnosis of OSA in children [42]. As an example, the European guideline accepts hospital RP as a valid alternative for the diagnosis of OSA in children and is considered an adequate screening technique when PSG is not available (Table 3).

### 1.6. Treatment

The goal of OSA treatment is complete resolution of SDB. This may require combining strategies (Figure 4), although the first-line treatment for OSA in children is adenotonsillar surgery [5] when adenotonsillar hypertrophy is present. Nonetheless, in the recent past, this treatment has been questioned. Recent publications have shown that the use of adenotonsillectomy in pediatric OSA patients may have variable results, reaching an AHI of 1 or less in about 50–70% of cases, but its efficacy decreases with risk factors such as age (<7 years), severe disease, chronic asthma or obesity. Persistent disease is present in 20–75% of children, with more than half having habitual snoring [7,43,44]. In addition, other surgical procedures may be performed in selected cases, such as septoplasty, uvulopharyngopalatoplasty, epiglottoplasty, glossopexy, and maxillomandibular surgery.

For those children with residual OSA following adenotonsillectomy or those in whom surgery is contraindicated or without adenotonsillar hypertrophy, positive airway pressure (PAP) therapies can be an effective treatment. The two types of PAP therapies prescribed in children to treat OSA are continuous positive airway pressure (CPAP) and bi-level positive airway pressure (Bi-PAP) [45]. CPAP is the most-commonly used PAP therapy, also recommended in children with craniofacial abnormalities or neuromuscular disorders [1,3,5]. The use of BiPAP is for patients intolerant to CPAP to treat nocturnal hypoventilation [46].

Positional therapy, as an alternative treatment, has been widely studied and relatively implemented in adults for the management of positional OSA. Positional OSA is defined when, spending more than 20% of sleep time in the supine position, the AHI in the supine position is at least double that in the non-supine position. This definition has not been adapted to the pediatric population and is, therefore, assumed in the child. In adults, positional therapy is incorporated in cases of mild–moderate OSA of positional origin and in those with severe OSA in order to lower CPAP pressure or when there is intolerance to first-line treatment. However, the indications in the pediatric population are not clearly established, and the scientific evidence is scarce. In this sense, it seems that children without tonsillar hypertrophy or with residual OSA could benefit from it, mainly in cases of obesity [47,48]. Therefore, randomized studies are necessary to establish the efficacy and indications of this type of therapy in the pediatric patient.

Weight management, orthodontic treatment, or medical therapy are offered as an alternative to surgery, especially in children with mild OSA [7,8,49] or when surgery is not indicated or contraindicated. There are data supporting that weight loss, if the child is overweight or obese, can improve OSA (hence, proposed to be considered first-line treatment in this population) [50]. Rapid maxillary expansion or orthodontic appliances are used to widen the palate and cause flattening of the palatal arch. On the other hand, medical therapies such as anti-inflammatory medications (nasal corticosteroid and/or oral montelukast) can also be used. There is little evidence about anti-inflammatory therapies in children. The results of randomized clinical trials evaluating the efficacy of intranasal corticosteroids for the treatment of OSA are not conclusive. Montelukast has short-term beneficial treatment effects for OSA in healthy, non-obese, surgically untreated children in terms of reducing the AHI, but the clinical relevance remains unclear [51,52]. Finally, myofunctional therapy has been accepted as a non-invasive treatment for OSA in children, as it may improve the AHI and oxygen saturation, at least after tonsillectomy or as an adjunct OSA treatment [53,54].

It is worth mentioning that obstructive SDB can be resolved spontaneously, particularly in children with mild OSA and adenotonsillar hypertrophy. Improvements may be due to the regression of lymphoid tissue or growth of the airway [55].

In summary, current data on the management of pediatric patients with OSA around the world, presented in this review, manifest important discrepancies and the need to be updated and homogenized. An agreed upon definition is needed with specific cutoff points to establish the diagnosis and levels of severity based on the associated risk and comorbidities. Clear diagnostic management algorithms must be settled upon in which it is defined when the simplified methods are useful in pediatric patients, in order to avoid underdiagnosis. It is necessary to identify prognostic markers that set up cutoff points for treatment indications based on objective impact. In addition, new metrics that better evaluate the disease could lead to the establishment of new protocols that improve the treatment management of the child. The objective is to explore new paradigms in the definition, diagnosis, and treatment of OSA in children. In this review, we analyzed the possible changes in the immediate future in this sense and what could be the new variables to establish the indication for the treatment of OSA.

## 2. Discussion

### 2.1. OSA Definition and Classification

While in adults, the definition of OSA is well established, this is not so clear in children, where these limits are between 1 and 3 events per hour, without an exact definition and important differences between guidelines. The new scientific evidence provided in recent years should be the basis for this new consensus and concrete definition. In the same way, the cut-off points for the levels of severity should be based on existing evidence about the associated risk related to the different levels of disease. In our opinion, an objective and concrete definition of OSA is needed, and the cutoff for severity levels should be stabilized based on the consequences of the illness.

### 2.2. OSA Diagnosis Algorithm

Related to the diagnostic management, PSG continues to be the gold standard test. Due to the scarcity of accredited sleep laboratories for children and important resources required to perform PSG, there has been a considerable effort to develop alternative diagnostic methods that are more widely available. However, more simplified tests have been developed, but insufficiently evaluated to be generally implemented in the pediatric population [56,57]. Hospital RP includes measures based on snoring, oronasal airflow, body position, chest movements, electrocardiography, and pulse oximeter saturation and has been used in Europe and Spain as a valid alternative to PSG [1,58]. Nonetheless, AASM’s position is that these tests may underdiagnose OSA [59] and are less implemented in pediatric OSA. However, RP in adults has been demonstrated to be a very useful tool for underdiagnosis control when used in the right population and with an adequate interpretation. There have been many suggestions of performing home sleep apnea tests (HSAT) [21] in children similarly to adults. However, the AASM position is that it is not recommended for the diagnosis of OSA in children [60].

Other modalities such as nocturnal oximetry recordings are a great possibility due to their low cost and their ease of use [61]. First, Brouillette and colleagues in 2000 [62] showed oximetric recordings as an approach for the identification of children with severe OSA. They concluded that a recording with at least three oxygen saturation (SpO_2_) drops to less than 90% and three clusters of desaturation events could be considered diagnostic for OSA [62,63]. Warapongmanupong and Preutthipan in 2019 [64] suggested another oximetry parameter, where calculating the dispersion of SpO_2_ could be useful in the initial investigation of OSA in children. A standard deviation of 1.06 or more could predict moderate–severe OSA with confidence.

Sleep questionnaires are helpful screening tools, but they cannot replace PSG [1]. To improve the diagnostic ability of clinical questionnaires, Villa et al. in 2013 [65] developed the Sleep Clinical Record (SCR), which included physical examination, patients’ subjective symptoms, and the presence of inattention and hyperactivity. This score was designed to detect children with SDB and AHI > 1 episode/h. A recent study showed that snoring children with an SCR score above 8.25 could identify those with moderate-to-severe OSA [66]. The European guideline accepts this score as a valid method to diagnose OSA and determine its severity when PSG is not available [1].

In our opinion, an update to OSA diagnostic management in children should be performed, based on the actual epidemiological situation and technological advancement. The level of clinical suspicion should be established, based on medical history, and the utility of different diagnostic methods should be provided. These and other simplified diagnostic methods have been successfully applied to adults and should be evaluated and incorporated into the management of children to avoid misdiagnoses.

### 2.3. Specific OSA Biomarkers

Besides, diagnosis and severity classification are limited to the AHI in both children and adults, although this sole parameter does not reflect the heterogeneity of the disease. Additionally, severity groups are arbitrary and do not include consequences. In addition to the AHI, it should be recommended to take into account the immediate consequences (intermittent hypoxia (IH)), symptoms, obesity, and comorbidities present in the child.

A new stratification of OSA, based on a phenotype identification and with prognosis implications, is necessary. This path towards personalized medicine has had its advances and applications in adults, but it has not been replicated in the same way in children. It is known that groups of patients with similar characteristics can benefit from specific diagnostic and therapeutic management [67].

The limitations of the AHI are mainly related to the complexity of the disease. Besides, short- and long-term consequences make it necessary to develop new parameters that improve disease detection, predict associated health problems, and provide a better response to the impact of treatment. Individuals with similar AHI levels differ in the patterns of hypoxia, cardiac autonomic response, and respiratory arousal intensity that drive CV disease [68], reflecting that the AHI may not fully characterize the physiologic disturbances of OSA. For this reason, there is an increasing effort to develop new parameters that include these aspects.

First, measuring pro-inflammatory biomarkers in blood or urine such as interleukin 6, tumor necrosis factor alpha, or CRP may be relevant for a better characterization of the disease and may be helpful in evaluating the changes produced with different treatments [69].

Secondly, heart rate variability (HRV) has been recently described as an OSA-specific biomarker, with promising results in evaluating treatment efficacy in children [70,71,72,73]. HRV corresponds to variations in the heart rate (HR) or beat-beat time interval due to the modulation of the autonomic nervous system’s activity. Because OSA sleep fragmentation and hypoxemia may increase the sympathetic nervous system’s activity, HRV analysis could evaluate these characteristic variations produced by OSA [72,74].

Thirdly, high BP levels, even values close to normal, are a well-known modifiable factor for developing high blood pressure (HBP) [75] and increased CV risk [76,77] in the future. Thus, measuring BP in children could be considered a good CV risk marker. The pathophysiology of HBP in OSA depends on various factors, apart from increased sympathetic tone, such as peripheral vasoconstriction, increased renin–angiotensin–aldosterone activity, and proinflammatory responses [40,78], causing persistent increases in vascular resistance and altering BP [79]. Related to BP, it has been demonstrated that OSA in adults is a risk factor for developing a nondipping profile [80,81], produced when nocturnal BP decreases less than 10% of daytime BP. Moreover, this nondipping pattern is described as an independent risk factor for the development of adverse CV events [82]. However, much less is known in children, where only some studies have assessed the impact of OSA in circadian BP patterns [83,84,85]. Controversial results on this issue have questioned if nocturnal dipping is truly preserved in OSA children. Therefore, more studies in children are needed in order to evaluate the utility of this marker in the prevention of adverse consequences in adulthood.

Fourthly, IH occurs as a result of OSA and is considered the main deleterious factor involved in the consequences for CV risk associated with OSA. Conventional measures of hypoxia such as the oxygen desaturation index (ODI) or the percentage of time during sleep with oxygen saturation below 90% (T90) are commonly used to characterize OSA and its relationship with CV risk. These parameters have shown a better prediction of CV risk and mortality than the AHI [86,87]. Nevertheless, some of them are non-specific for OSA, reinforcing the idea of developing and validating more quantitative hypoxia metrics. In this sense, a novel OSA-specific biomarker, hypoxic burden (HB), has emerged for a better characterization of OSA, which focuses on the frequency, duration, and depth of the respiratory events. HB is defined as the total area under the desaturation curve related to the respiratory event. The value is obtained by adding the individual desaturation areas and dividing the total area by the duration of sleep, in units of %min/h [88]. This parameter has been shown to be significantly associated with adverse CV outcomes in recent studies in adults [89]; however, the search within this issue has not yet been evaluated in children. Measuring HB in children with OSA may be important and of interest, as it could better predict the risk attributed to the disease and could improve the choice of treatment. Besides, HB measurement requires only recording of airflow and oxygen saturation signals, so it could be easily obtained by simplified methods, facilitating its implementation in clinical practice.

Therefore, an effort to phenotype and establish the value of the different biomarkers on the way to personalized medicine is needed in the OSA pediatric population.

### 2.4. Treatment

In reference to treatment, the lack of adenotonsillectomy effectiveness in some groups of patients reflects that they may have a different response to treatment and/or a different prognosis. Furthermore, OSA treatments are not exclusive, and all the therapeutic options should be considered to approach treatment individually. For this, we highlight the importance of looking for the reversible cause of the disease, so that future risk can be prevented. The most-common cause is adenotonsillar hypertrophy, but other risk factors such as obesity and craniofacial dysmorphism should be taken into account for the correct management of the patient. In this sense, the treatment of obesity in adult patients is a cornerstone in its control, becoming ahead of CPAP treatment in overweight/obese patients in the latest consensus document [90]. It is true that the presence of obesity in children is less prevalent, but there has been a significant increase in recent times. This fact contributes to the higher frequency of OSA cases in the absence of tonsillar hypertrophy and residual OSA after tonsillectomy due to overweight/obesity. However, the impact of obesity treatment in children has not been studied to the same extent as in adults, nor has its role in the diagnostic algorithm been established. Other different possibilities that could be used include the use of CPAP or orthodontic treatment in patients with certain craniofacial abnormalities. The early use of these techniques could prevent the risk of suffering from OSA.

On the other hand, the development of innovative OSA parameters related to CV risk in the pediatric population could initiate new paradigms in the management of children with OSA, as they could better predict and determine which patients would benefit from treatments. One possibility could be measuring the HB in the pediatric population as a substitute of the AHI in the diagnosis and classification of the severity groups of OSA. Relating this parameter to BP values could indicate which groups would benefit from OSA treatment in terms of the prevention of future CV risk.

Additionally, the reevaluation of children should be performed after treatment to determine whether further treatment is necessary, especially in children at risk of residual OSA. It is not clear when the ideal moment for this reassessment is, specific recommendations based on different available therapies being necessary.

In summary, OSA treatment in children should focus on considering these two important aspects: reversible causes of the disease and measurements defining which patients would benefit from them. New clinical trials would be needed to assess the feasibility of the implementation in clinical practice of these innovative insights of the management of OSA in children.

## 3. Conclusions

OSA in the pediatric population has important limitations that should be updated.

First, a specific and consensus definition should be established. Second, the role of simplified methods in the diagnosis algorithm should be considered. Third, new biomarkers for risk identification are needed in this population. Finally, personalized medicine should be implemented in this population.

## Figures and Tables

**Figure 1 biomedicines-11-01708-f001:**
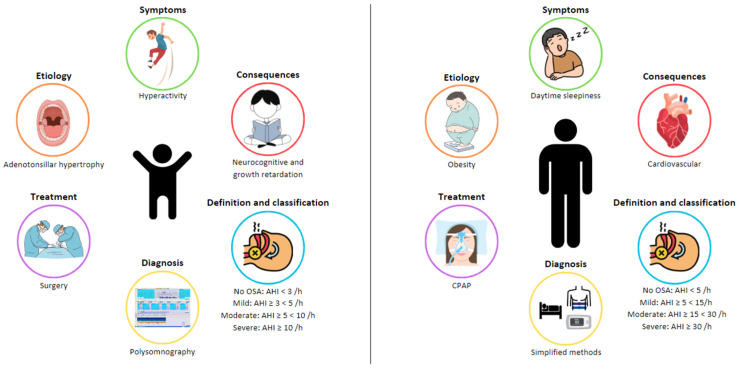
Differences in obstructive sleep apnea (OSA) between adults and children.

**Figure 2 biomedicines-11-01708-f002:**
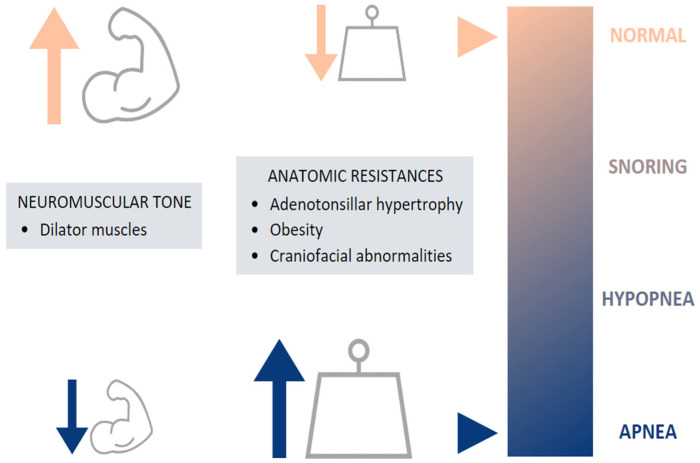
Etiology of pediatric OSA.

**Figure 3 biomedicines-11-01708-f003:**
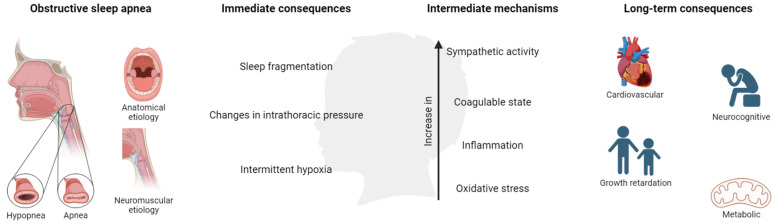
Mechanisms and consequences of OSA in children. Created with BioRender.com. This population is characterized by poor academic performance showing a reduction in memory capacities and difficulties in learning and attention (especially in specific areas such as mathematics, science, reading, and spelling) [24,25,26], which could be associated with hyperactive behavior during the day.

**Figure 4 biomedicines-11-01708-f004:**
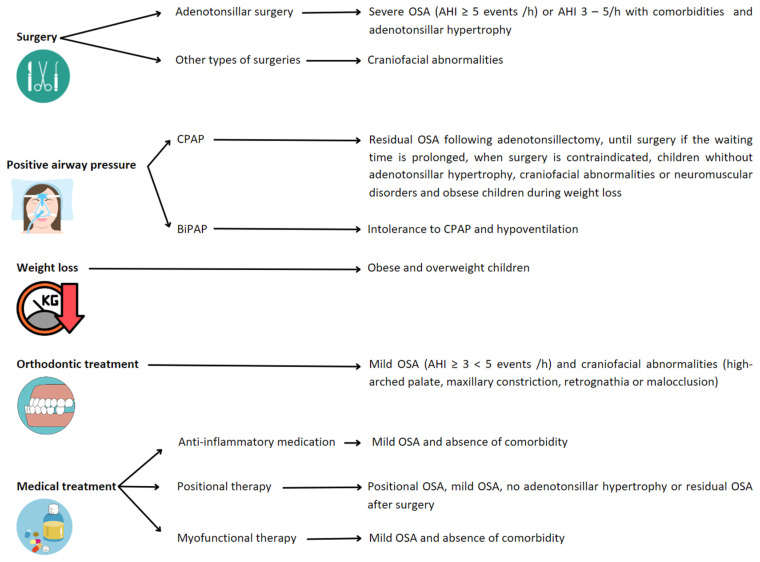
Recommended treatments depending on conditions leading to OSA in children. Abbreviations: OSA: obstructive sleep apnea; AHI: apnea/hypopnea index; CPAP: continuous positive airway pressure; BiPAP: bi-level positive airway pressure.

**Table 1 biomedicines-11-01708-t001:** Symptoms of OSA in children.

Nocturnal Symptoms	Daytime Symptoms
Snoring	Behavioral disorders
Witnessed apneas	Neurocognitive disorders
Gasping	Mood instability
Oral breathing	Excessive daytime sleepiness
Paradoxical thoracic movements	
Nightmares	
Restless sleep	
Nocturnal enuresis	

**Table 2 biomedicines-11-01708-t002:** Consequences of OSA in children.

Author (Year)	Number of Participants	Age (Years)	OSA Severity Criteria	Outcomes	Results
Behavioral and neurocognitive sphere					
Menzies et al., 2022 [38] Metanalysis of 63 studies	17,834	From 2 to 18 years	Due to the lack of a consensus severity criterion, the subgroup given by the author was used (e.g., mild OSA)	Intelligence, attention, memory, visual spatial skills, and language	Children with SDB had significant impairments in all cognitive domains, intelligence being the most-affected quality. These neurocognitive deficits were found in primary snorers among OSA children.
Growth retardation and metabolism					
Lagravère et al., 2019 [39] Systematic review of 12 studies				Growth mediators (IGF-I and IGFBP-3)	Children with OSA present lower levels of growth mediators, indicating growth retardation, significantly higher cardiovascular disease risk, and decreased cognitive functions compared to healthy controls. Tonsillectomy may improve all these functions with a great impact on general health.
Cardiovascular sphere					
Ai et al., 2022 [40] Metanalysis of 14 studies	3081	3 to 17 years	Mild OSA is defined as an AHI between 1 and 5 events per hour Moderate to severe is defined as an AHI ≥ 5 /h.	BP parameters: awake and nighttime SBP and DBP	The mean SBP was higher in children with mild or moderate-to-severe OSA compared to healthy controls, these effects being more pronounced during the night. The results suggest that moderate-to-severe OSA in children is associated with a higher risk of adverse SBP outcomes.

Abbreviations: OSA: obstructive sleep apnea; SDB: sleep-disordered breathing; IGF-1: insulin-like growth factor 1; IGFBP3: insulin-like growth factor-binding protein 3; AHI: apnea/hypopnea index; BP: blood pressure; SBP: systolic blood pressure; DBP: diastolic blood pressure.

**Table 3 biomedicines-11-01708-t003:** Diagnosis and management of OSA in children.

Guide	Diagnosis and Management of OSA in Children
Spanish Society of Pneumology and Thoracic Surgery (SEPAR)	This guide divides the OSA diagnostic methodology between primary care and hospital care in order to increase the diagnostic efficiency. In primary care, the evaluation of the child with suspected OSA (presence of snoring and symptoms or suggestive clinical findings) should include the medical history and complete clinical examination. Medical history: family history, events related to the child’s sleep and breathing, and sleep questionnaire (Chervin). Complete clinical examination: craniofacial and UA anatomy, cardiopulmonary examination and somatometry. Children with obesity represent a special risk group. Depending on the results, referral of the patient from primary care to the reference sleep unit is considered. If there is suspected OSA in the clinical history and or/Chervin, retrognathia, adenotonsillar hypertrophy and Mallampati ≥ 2, hospital RP is performed. Otherwise, a control visit is carried out 6 months after baseline visit. When the index of respiratory events is ≥ 5 in the RP, children are referred to adenotonsillar surgery. With an inferior result, a PSG is performed. An AHI ≥ 5/h on PSG leads to adenotonsillar surgery. If the AHI is < 3/h, an anti-inflammatory therapy or review visit after 6 months is assessed. For an AHI 3–5/h comorbidities are evaluated. The presence of comorbidity leads to adenotonsillectomy and anti-inflammatory therapy is selected when there is absence of comorbidity. All children should be clinically reassessed after surgery (recommended in the next 6 months), performing a sleep study in children with severe preoperative OSA or when risk factors or OSA symptoms persist, where other treatments such as diet, CPAP, or orthodontics will be assessed.
European Respiratory Society	The diagnosis and management for SDB is described as a stepwise approach in 7 steps. Identification of risk of SDB: symptoms of UA obstruction, alterations in physical exam, objective findings related to SDB and/or prematurity or family history of SDB. Identification of comorbidities in CV system, CNS, nocturnal enuresis, growth delay or decreased QoL and conditions coexisting with SDB such as recurrent otitis media and history of tympanostomy tube placement, wheezing or asthma, metabolic syndrome or oral-motor dysfunction. Recognition of factors predicting long-term persistence of SDB: obesity, male sex, obstructive AHI > 5/h, African-American ethnicity and persistent tonsillar hypertrophy and narrow mandible. Objective diagnosis and assessment of SDB severity: PSG or RP is indicated in children at risk of SDB. (1) OSA definition 1: obstructive AHI ≥ 2/h or obstructive apnea index ≥ 1/h with SDB symptoms; (2) OSA definition 2: SDB symptoms and AHI ≥ 1/h. No alternative methods can substitute PSG but could be used in low resource settings: ambulatory PSG or RP, nocturnal oximetry, Pediatric Sleep Questionnaire or Sleep Clinical Record. Indications for treatment of SDB: indicated when AHI > 5/h. When PSG or RP are not available, treatment is considered when positive oximetry or SDB questionnaires or morbidity is present. It is unclear whether should treat primary snoring (evaluation annually). Stepwise treatment approach for SDB is usually implemented until complete resolution of SDB: (1) weight loss in overweight and obese children; (2) nasal corticosteroids and/or montelukast in non-obese and < 6 years children; (3) adenotonsillectomy in children with OSA and adenotonsillar hypertrophy; (4) rapid maxillary expansion or orthodontic appliances in children with OSA and maxillary constriction, retrognathia or malocclusion; (5) CPAP or NPPV when residual OSA after adenotonsillectomy or hypoventilation; (6) craniofacial surgery when syndromic craniofacial abnormalities; (7) tracheostomy in severe OSA when other nonsurgical or surgical interventions have failed or are contraindicated. Recognition and management of persistent SDB: outcomes monitored after intervention are: symptoms, PSG (or RP, oximetry/capnography when not available), QoL, CV or CNS morbidity, enuresis and growth rate. PSG or RP should be performed, between 6–12 weeks after treatment, in children at risk of persistent OSA, after adenotonsillectomy, in children with persistent symptoms or children with mild OSA treated with corticosteroids and/or montelukast. PSG should be performed 12 months after rapid maxillary expansion and after 6 months when oral appliance treatment is selected. At least, one PSG or RP annually should be used to titrate CPAP or NPPV.
American Academy of Pediatrics	This practice guideline focuses on uncomplicated childhood OSA, associated with adenotonsillar hypertrophy and/or obesity in an otherwise child who is being treated in the primary care setting. It comprises 8 key action statements. Screening for OSA. If the child presents signs or symptoms of OSA, clinicians should perform medical history and physical examination. Snoring and findings in the evaluation should lead to PSG (gold standard test) or alternative tests when PSG is not available (nocturnal video recording, nocturnal oximetry, daytime nap PSG or ambulatory PSG). Adenotonsillectomy is recommended when the child is determined to have OSA and adenotonsillar hypertrophy (and do not have contraindication to surgery). If the child has OSA but not adenotonsillar hypertrophy other treatment should be considered. Monitoring of high-risk patients undergoing adenotonsillectomy. Reevaluation. Clinical reassessment should be performed in all patients with OSA for persisting symptoms after therapy to determine whether further treatment is required (6 to 8 weeks after treatment). Clinicians should refer patients for CPAP management if symptoms persist after adenotonsillectomy or if it is not performed. Weight loss is recommended in addition to other therapy if the child with OSA is overweight or obese. Intranasal corticosteroids may be prescribed for children with mild OSA in whom surgery is contraindicated or have mild postoperative OSA (<5/h).

Abbreviations: OSA: obstructive sleep apnea; UA: upper airway; RP: respiratory polygraphy; PSG: polysomnography; AHI: apnea/hypopnea index; CPAP: continuous positive airway pressure; SDB: sleep-disordered breathing; CV: cardiovascular; CNS: central nervous system; QoL: quality of life; NPPV: non-invasive positive pressure ventilation.

## Data Availability

Not applicable.

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
