# Peer review of "Diagnosis and Treatment of Sleep Apnea in Children: A Future Perspective Is Needed"

_biomedicines, 2023, doi:10.3390/biomedicines11061708_

Round 1
Reviewer 1 Report
This paper provides a significant review about OSA and points out that OSA in the pediatric population has important limitations that should be renovated. First, a specific and consensus definition should be established. Second, the role of simplified methods in the diagnosis algorithm should be considered. Third, new biomarkers for risk identification are needed in this population. Finally, personalized medicine should be implemented in this population. These important insights into diagnosis and treatment of sleep apnea in children are indeed critical for the relevant future research, so I think this review can be publishable in this journal after addressing some minor issures as follows:
1. More figures shoud be provided so as to make this paper more readable and understandable.
2. Some statistic data should be added to offer the evidences for some descriptions and discussions especially for the consequences, diagnosis and treatment.
Author Response
Answer to reviewer 1:
Thank you to the reviewers for the comments that have improve our manuscript. In the next letter we give answer to the suggestions point by point:
- More figures should be provided so as to make this paper more readable and understandable.
New figures and tables have been added to the main text.
- Some statistic data should be added to offer the evidences for some descriptions and discussions especially for the consequences, diagnosis and treatment.
A brief literature review has been done in order to show specific evidence for the consequences and statistics data. Discusion about the diagnosis and treatment have been enlarged too.
Reviewer 2 Report
The article is relevant to OSA research and as such is of interest to readers of Biomedicines. The manuscript is well written and the flow is good. The abstract adequately describes the study.
I noticed some minor issues:
1) Are there data on positional therapy in the pediatric population?
2) Please discuss studies on the effectiveness of weight loss in pediatric polulation.
Accordingly, in my opinion the papaer should be published in Biomedicines.
Author Response
Answer to reviewer 2:
Thank you to the reviewers for the comments that have improve our manuscript. In the next letter we give answer to the suggestions point by point:
- Are there data on positional therapy in the pediatric population?
There is few evidence about positional therapy in the paediatric population. Recent studies have been added to the manuscript evaluating positional devices in obese and overweight children, highlighting the potential use of this therapy.
- Please discuss studies on the effectiveness of weight loss in pediatric population.
A brief discussion of weight loss effectiveness has been added.
Reviewer 3 Report
This manuscript (biomedicines-2433491) was well-written and described recent trends focused on the diagnosis and treatment of pediatric sleep apnea. I have a few recommendations for this manuscript.
1. Line 56-57: In Figure 1, there doesn’t seem to be a distinct entity with a very different profiles from adults. Etiology, immediate consequences, and long-term consequences are similar to adults, except for growth retardation.
2. In Figure 1, I recommend insert intermediate mechanisms after immediate consequences, i.e., Obstructive sleep apnea à immediate consequences à intermediate mechanisms à long-term consequences. Kim’s paper will be helpful (Kim JK et al. Respiratory Physiology & Neurobiology 2011:178:465-474).
3. Compared to the diagnosis part, the treatment part seems to less covered.
3-1. Please describe other PAP treatment in children such as BiPAP, in addition to CPAP.
3-2. Detailed research findings or indications for other treatment options for children, such as anti-inflammatory medications, positional therapy, and myofunctional therapy, should be presented.

Author Response
Answer to reviewer 3:
Thank you to the reviewers for the comments that have improve our manuscript. In the next letter we give answer to the suggestions point by point:
- Line 56-57: In Figure 1, there doesn’t seem to be a distinct entity with a very different profiles from adults. Etiology, immediate consequences, and long-term consequences are similar to adults, except for growth retardation.
The differences between adults and children have been more clearly explained and synthesized in the new Figure 1.
- In Figure 1, I recommend insert intermediate mechanisms after immediate consequences, i.e., Obstructive sleep apnea à immediate consequences à intermediate mechanisms à long-term consequences. Kim’s paper will be helpful (Kim JK et al. Respiratory Physiology & Neurobiology 2011:178:465-474).
Intermediate mechanisms have been included in Figure 3 as suggested.
- Compared to the diagnosis part, the treatment part seems to less covered.
3-1. Please describe other PAP treatment in children such as BiPAP, in addition to CPAP.
3-2. Detailed research findings or indications for other treatment options for children, such as anti-inflammatory medications, positional therapy, and myofunctional therapy, should be presented.
Treatment part has been enlarged and PAP, BiPAP, antiinflammatory, positional and myofunctional therapies have been included.
Round 2
Reviewer 3 Report
The manuscript has been well modified according to reviewer's recommendations. Thank you for the author's efforts.